# Why involve families in acute mental healthcare? A collaborative conceptual review

Aysegul Dirik,[1] Sima Sandhu,[1] Domenico Giacco,[1,2] Katherine Barrett,[1] Gerry Bennison,[1] Sue Collinson,[1] Stefan Priebe[1]

[1]Unit for Social and Community Psychiatry, WHO Collaborating Centre for Mental Health Services Development, Queen Mary University of London, London, UK
[2]East London NHS Foundation Trust, London, UK

**Correspondence to**
Aysegul Dirik;
a.dirik@qmul.ac.uk

## ABSTRACT

**Objectives** Family involvement is strongly recommended in clinical guidelines but suffers from poor implementation. To explore this topic at a conceptual level, a multidisciplinary review team including academics, clinicians and individuals with lived experience undertook a review to explore the theoretical background of family involvement models in acute mental health treatment and how this relates to their delivery.

**Design** A conceptual review was undertaken, including a systematic search and narrative synthesis. Included family models were mapped onto the most commonly referenced underlying theories: the diathesis–stress model, systems theories and postmodern theories of mental health. Common components of the models were summarised and compared. Lastly, a thematic analysis was undertaken to explore the role of patients and families in the delivery of the approaches.

**Setting** General adult acute mental health treatment.

**Results** Six distinct family involvement models were identified: Calgary Family Assessment and Intervention Models, ERIC (Equipe Rapide d'Intervention de Crise), Family Psychoeducation Models, Family Systems Approach, Open Dialogue and the Somerset Model. Findings indicated that despite wide variation in the theoretical models underlying family involvement models, there were many commonalities in their components, such as a focus on communication, language use and joint decision-making. Thematic analysis of the role of patients and families identified several issues for implementation. This included potential harms that could emerge during delivery of the models, such as imposing linear 'patient–carer' relationships and the risk of perceived coercion.

**Conclusions** We conclude that future staff training may benefit from discussing the chosen family involvement model within the context of other theories of mental health. This may help to clarify the underlying purpose of family involvement and address the diverse needs and world views of patients, families and professionals in acute settings.

## BACKGROUND

Practically, all mental health policies and guidelines suggest some form of family, friend or carer involvement in patients' mental healthcare (hereon abbreviated to 'family involvement'). The potential benefits

### Strengths and limitations of this study

► We included distinct family involvement models used internationally in acute mental health treatment.
► We explored the role of underlying theories and potential risks of harm, both of which may impact implementation.
► Our multidisciplinary team included the active contribution of people with lived experience of acute mental health treatment as well as clinicians and academics.
► Broadly mapping across models means we did not include an exhaustive list of every single variation of family involvement in acute treatment.
► The identified models were originally developed in various Western mental health settings, which might not reflect the theoretical frameworks of non-Western settings.

of this for patients are well documented, including relapse prevention and reduced hospital stays.[1 2] Despite growing consensus in policy toward family-inclusive services, in reality, audits consistently highlight poor implementation rates.[3] This problem is well documented: over decades of research, frustrations have been expressed about the difficulties of implementing family involvement into routine psychiatric care.[4 5]

To complicate matters, the reason for conducting family involvement in the first place cannot be traced to a single school of thought or point in time. Sociopolitical events, such as the deinstitutionalisation of mental health services and early theories of mental illness have meant that families often felt both blamed for mental health problems as well as being given the responsibility of providing support. Family advocacy groups have pushed for policy changes towards recognising the support that families provide, and the burden that can be associated with this.[6] Alongside this, multiple family involvement

models have emerged based on divergent theories of the nature of mental health problems.

The use of family involvement models can vary highly between services.[7][8] Evidence is emerging of a lack of shared understanding of what constitutes appropriate family involvement and how to best incorporate it into services.[9] Such discord is problematic, as it can impact staff attitudes and the general organisational culture toward family work.[10] This, in turn, has implications for resource allocation and intervention delivery, particularly if there is disagreement about the aim or value of conducting it.[11][12]

A recent review by members of our team identified multiple barriers to the implementation of family involvement at the individual, team and organisational level.[13] These barriers were common across intervention models and international settings. A particularly challenging setting is acute treatment, which typically involves admission to hospital for inpatient treatment or a crisis intervention in the community. Clinician reports indicate numerous difficulties in implementing family involvement in these contexts, which are often characterised by a strong focus on risk reduction and crisis management.[14]

Revisiting the concepts underlying family work seems timely as it may bring us a step closer to understanding how to implement it in a way that is in keeping with the values of mental health organisations, users of their services and families. This review seeks to explore the diversity across different family involvement models and to consider how their underlying theoretical backgrounds might impact on how they are delivered and received today. We investigated the following questions:

1. Which family involvement models are used in general adult acute mental health settings?
2. What is the theory or rationale underlying family involvement models?
3. What are the components of the models?
4. What is the role of patients and family in the delivery of the models?

## METHOD

For this review, we did not aim to produce an exhaustive list of every existing family involvement model. Instead, we set out to find distinct approaches that represented the diversity of the models that are used today. A conceptual review,[15] which enables the exploration of the breadth of concepts in a given area, was considered the most appropriate methodology to answer these research questions.[16–19] This review was preregistered on PROSPERO (CRD42016032749).

### Search strategy and selection criteria

A wide search strategy was employed, including a systematic search of electronic databases (Embase, MEDLINE, PsycINFO, BNI, CINAHL and AMED) for descriptors of 'family/carers' 'mental health' 'model/approach' and 'setting' and hand searches (see online supplementary appendix 1). AD conducted the searches in consultation with SS and experts in the field. As described below, the searches and analyses were iterative and the chosen key models were finalised during the first stage of analysis.

We included (1) key texts containing an original description of a family involvement model that (2) referred to the management of an acute mental health situation or the treatment of 'severe mental illness' that could be started during the acute phase (3) with a clear description of how families are involved in the patient's treatment and (4) the primary focus was general adult mental health (ages 18–65).

Papers were excluded if (1) the word 'carer' was being used to refer to paid staff members, (2) the primary focus was on specialist services, (3) they were a description of a family therapy model rather than a programme designed for family involvement in acute mental health treatment or (4) it was not possible to obtain an English-language description, although non-English texts were translated whenever possible.

The criteria meant that we could only include models where the primary focus was to involve family members in order to support the patient's care in acute settings. While other approaches to involving families exist, they were considered to be beyond the scope of the current review.

### Data analysis

A multidisciplinary review team was formed to minimise biases in the searches and analyses.[15] This composed of the lead researcher (AD, a doctoral researcher), a research psychologist (SS), two clinical/academic psychiatrists (DG and SP), three individuals with lived experience of acute mental health treatment, either as patients or as family members (KB, GB and SC) and a clinical nurse manager, who also has research experience (PM). The review team worked most closely on steps 2 and 3 below.

A narrative synthesis was conducted to reach a thorough conceptual understanding of family involvement models.[20] The steps described below were highly iterative:

1. To develop a preliminary synthesis, found texts were clustered into categories of family involvement models. The clustered groups were expanded and collapsed until the family involvement models were broadly similar within each group and sufficiently different from the other groups. Then, the key texts were identified within each group by reference screening, citation checking and snowballing to find original descriptions of the approach. The final inclusion decision was made after discussions with colleagues and experts in the field. Alongside this, the theoretical references of the models were identified by extracting the change processes and reasons for intervention development described in the texts, as well as reference screening, reading widely around the subject area and consulting experts. The chosen family involvement models were mapped on to the identified theories.

2. Components of the included studies were identified by extracting authors' descriptions of how the model is carried out and clustering the text into similar methods. Similarities and differences were then compared across the models.

3. We explored the role of patients and carers within and between the models using thematic analysis.[21] A selection of family involvement models was examined in depth, ensuring that there was representation across the identified theoretical references. Analysis of the emerging key themes was conducted iteratively after multiple reflective discussions.

4. As well as using our own multidisciplinary review teams, several measures were taken to ensure robustness of the synthesis. This included numerous consultations with a wider team of around 30 researchers, an expert in Family Intervention, two service user research groups (SUGAR, the Service User and carer Group Advising on Research, and SURF, the North London Service User Research Forum), and a Social Psychiatry expert academics meeting. The ROBIS

(Risk Of Bias In Systematic reviews) tool was used to guide our methodology, although quality assessment of the included studies themselves was not considered appropriate for this review of concepts.[22]

## RESULTS

The analysis was built up at each stage from (1) identifying distinct models, (2) mapping the models to their theoretical references, (3) comparing the model components and (4) exploring the role of patients and carers in the delivery of the approaches.

### Key models

We identified six distinct family involvement models from 16 key sources (table 1). This included four key family psychoeducation models, which were clustered due to the similarity of their underlying approach. The PRISMA flow diagram (figure 1) depicts our selection process.[23]

As will be described, some of the models (eg, family psychoeducation) were originally developed for the

| Model | Country | Description |
|---|---|---|
| Calgary Family Assessment and Intervention Models[40] | Canada | Guidelines for family nursing practice and assessment that draw on systems, communication and change theory. In acute care, interventions may target cognitive, affective and behavioural domains of family functioning to invoke change. Staff are trained to use systemic tools such as genograms for the assessment of social relationships. |
| ERIC (Equipe Rapide d'Intervention de Crise)[36] | France | Nurses, doctors and psychologists work together as a large multidisciplinary team in a mobile service. Brief psychotherapy is provided, usually in the patient's home, with the aim of 'enveloping' (containing) the crisis. There is strong emphasis on the role of communication and the competence of the family unit to deal with future crises. |
| Family Psychoeducation Models[33–35 43–45] | UK, USA | The most widely used model globally, developed from research into the role of family communication in relapse. Specialist teams provide a package of support including at least (1) an educational component about the patient's diagnosis and the recommended treatment; (2) problem-solving and/or communication training to simplify communication for the patient and (3) emotional support for the family. |
| Family Systems Approach, SYMPA (systems therapy methods in acute psychiatry)[30 31 37] | Germany | All staff across disciplines are trained to assess and treat patients within a systemic framework. This includes changing language use to less-medicalised terms. Staff are also trained as 'negotiators' between the patient and the organisation about matters such as medication and compulsory measures. |
| Open Dialogue[32 60 61] | Finland | A multidisciplinary mobile crisis team attend the patient's home within a short time from referral. Meetings including the patient's wider social network take place daily, and continue until a 'joint understanding' is reached of the patient's distress. The process of listening and responding is considered central in reducing the patient's distressed state. |
| Somerset Model[38 39] | UK | Service-wide approach, developed to address policy and advocacy-led calls for more family-inclusive services. All families are offered an initial needs assessment and information about the service and may be referred to more intensive provision. |

**Table 1** Family involvement models in acute mental healthcare

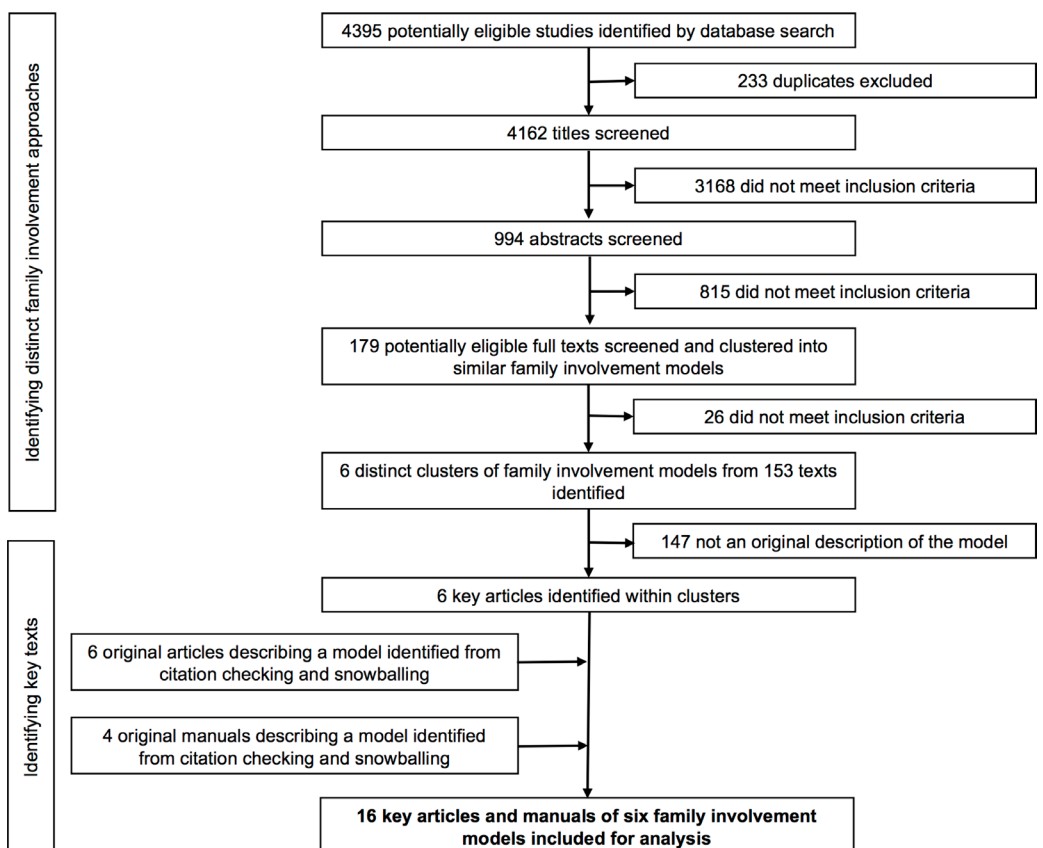

**Figure 1**  PRISMA flow diagram depicting study selection.

treatment of schizophrenia or psychosis and then adapted to be used more widely whereas others were less focused on a particular service context or diagnostic category.

### Theory mapping

We then explored how the models relate to their most commonly cited theoretical influences, which included systems theory, the diathesis–stress model and postmodern theories. There were some overlaps in the theoretical references underpinning the models, as illustrated in figure 2.

### Systems theories

Systems theories (eg, General Systems Theory and Cybernetics)[24–26] are commonly used frameworks for broadly understanding how all systems function and the importance of interactions in those systems. In psychiatry, a major application of these theories has been systemic family therapy.[27–29] Here, the broad principles posit that there is an issue within the family system and that one person within this becomes the designated 'patient' presented to services. The professional's role is to work with the whole family to influence the processes that contribute to the patient's mental state.

A minority of the examined models offer traditional systemic family therapy as a supplementary, intensive service for particular families.[30–32] However, the general influence of systems theories is substantial across the models. Historically, while some models were developed

in part as a reaction to the perceived 'blaming' attitude of systemic family therapy,[33–35] others used systemic principles as a guideline to set up family-inclusive services.[30 31 36–39] Notably, the majority of models use systemic techniques in their everyday practice, such as constructing genograms

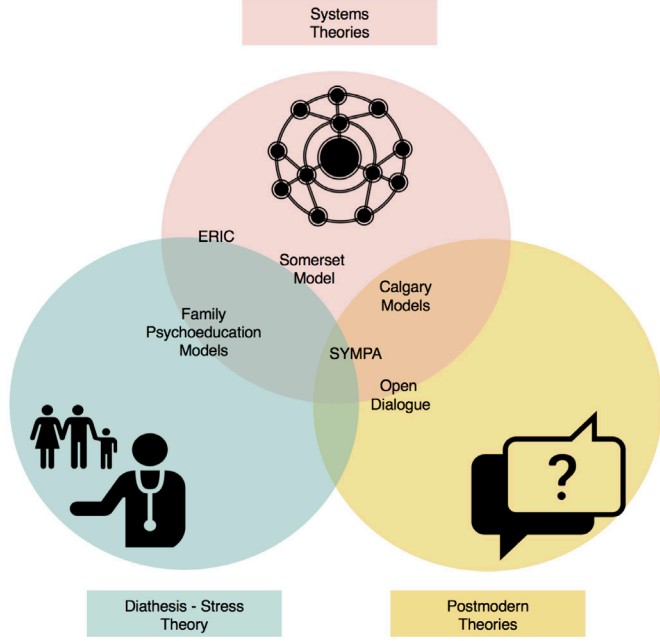

**Figure 2**  Placement of family involvement models within the diathesis–stress, systems and postmodern theories.

(social network diagrams) to understand the patient's social environment[30 38–40] and using therapeutic techniques such as circular questioning.[34]

### The diathesis–stress theory

The diathesis–stress model posits that people with schizophrenia have an underlying physical vulnerability to developing symptoms, the risk of which only manifests if the person experiences excessive environmental stress.[41] Research into potential environmental stressors highlighted several concepts of harmful (or helpful) communication patterns within families.[42]

Based on this theory, multiple family involvement models were developed to address the underlying biological causes of schizophrenia by using medication as well as providing support and 'psychoeducation' to families to teach them the diathesis–stress model and reduce stressful communication that could exacerbate symptoms.[33–35 43–45]

One commonly accepted theory of communication is 'expressed emotion' (EE), which suggests that levels of *hostility, criticism, overinvolvement* and *warmth* can affect a 'person's likelihood of relapse').[42 46] While some models intervened directly in EE communication,[34 43] others focused more on developing general communication skills within the family unit to encourage expression and improve problem-solving.[33]

Later, psychoeducation models were developed to include other mental and physical health conditions[47–50] and alternative modes of delivery, including mobile crisis teams[51] and online interventions, including those focused on family members' own needs.[52 53] In addition, some services have incorporated other concepts such as recovery and peer support.[54]

While the clearest influence of the diathesis–stress theory is on family psychoeducation models,[33–35 43–45] arguably its elements exist in all models that take place within a medical system. For example, almost all of the models routinely recommend medication along with psychological support and social interventions.[30 31 37].

### Postmodern theories

Postmodern approaches to mental health are often critical of commonly accepted narratives of 'mental illness'.[55] Influential theories within this sphere include social constructionism and constructivism, which broadly posit that mental health problems only exist in social contexts and so their solutions can only emerge within those contexts.[56–58] There are some overlaps with systemic theories, particularly in terms of the importance of viewing the patient within their social context. An area of difference is less focus on interactions within the 'system' and more on the individual perspectives people have on their own problems. As a result, postmodern approaches tend to focus less on particular interventions and more on working with uncertainty during the therapeutic process. While the rise in popularity in the 1960s is well documented,[56 57 59] these viewpoints were not translated to widely used family involvement models and largely fell out of favour for more medically focused approaches. However, some models developed from postmodern and systemic ideas, often as a reaction to more medical theories.[30–32 37 40 60 61]

While pure postmodernism rejects biological explanations, these models took a more integrated approach, incorporating postmodern theories and practices[62–64] into existing medical systems. Common features included strong emphasis on the wider social context[32 60 61] and prominence given to individual narratives and explanations[32 40 60] rather than emphasising diagnoses or highly structured treatment models, which could be viewed as imposing 'absolute truths' or world views.

### Synthesis of the components of the approaches

Considering the rich and divergent theoretical background of family involvement models, we examined how these theories related to the components of the models. This section refers to original descriptions of key family involvement models. In current practice, models have been adapted in countless ways to include different concepts. This includes variation to the methods of service delivery and the inclusion (or exclusion) of particular therapeutic components. However, the most common components across the original descriptions are summarised in table 2 and the following section.

### Communication/language use

Communication and language use were strongly emphasised across all models. Models adhering to the diathesis–stress theory intervened in EE or other aspects of communication as a tool in relapse prevention.[42 46] In Open Dialogue and ERIC (Equipe Rapide d'Intervention de Crise), while the theoretical basis was very different, the dialogue between and within participants was theorised as the main driver of change.[32 36 60 61] Systemic approaches such as the Calgary model trained staff in communication to improve families' service experience,[40] while in SYMPA staff were trained to avoid diagnostic labels, as they could disempower patients and encourage a 'psychiatric career' identity.[30 31]

### Joint decision-making and the role of experts

All the models emphasised the need to make decisions jointly, although the emphasis on experts differed. Models based on the diathesis–stress theory, which are more medical in nature, emphasised the importance of experts who provide *'information, advice and guidance'*.[34] However, this approach existed in other models, but was not acknowledged as openly. Other models *'incompatible'* with *'the illness concept inherent in the idea of vulnerability and the strong focus on compliance with psychopharmacological treatments'* (p377[31]) still described how clinicians needed to *'negotiate'* various aspects of the treatment with the patient and their family.[30 31 37] Postmodern-influenced models minimised professionals' role in treatment decisions and made all clinical decisions jointly with the patient and their wider social network. This often meant that the

**Table 2** Common components of family involvement models

| | Communication/language use | Joint decision-making | Support for the family | Wider social network | Medication use | Specialist teams/staff | Whole system approach |
|---|---|---|---|---|---|---|---|
| Calgary Family Assessment and Intervention Model[40] | √√ | √√ | √√ | √ | √ | – | √√ |
| ERIC (Equipe Rapide d'Intervention de Crise)[36] | √√ | √ | √√ | √ | √ | – | √√ |
| Family Psychoeducation Models[33–35 43–45] | √√ | √ | √√ | – | √√ | √√ | – |
| Family Systems Approach (SYMPA)[30 31 37] | √√ | √√ | √ | √ | √ | – | √√ |
| Open Dialogue[32 60 61] | √√ | √√ | √ | √√ | √ | – | √√ |
| Somerset Service Model[38 39] | √ | √ | √√ | √ | √ | – | √√ |

Key: √√, Strongly emphasised in model; √, Present in the model, not a key feature; −, Relatively less or no emphasis.

professionals had to 'tolerate uncertainty' in the treatment process.[32 60 61]

### Support for the family themselves

Some models were developed directly in response to families' stated needs for support and involvement.[34 38 39 43] This could, for example, prevent the build-up of problems which could manifest as poor communication.[43] More systemic or postmodern-influenced models were less focused on family support, and rather saw the involvement of the patient's wider social network as a necessary tool in understanding their social context.[30 31 37]

### Wider social network involvement

The involved 'family' differed across the models. Psychoeducation models generally focused on the people the patient lives with (and therefore interacts with the most), meaning it was largely aimed towards parents and partners.[33–35 43–45] This was sometimes a deliberate decision based on previous lack of utility when involving extended relatives and friends.[35] While some systemic models also referred to the family in this context,[36] others used the term family more widely to refer to any members of the patient's 'problem-determined system'.[37] Open Dialogue, which focuses heavily on interactions within the wider social network placed importance on all its members, including friends, family, neighbours and colleagues.[32 60 61]

### Medication use

Medication use featured in all models. Approaches that derive more from the diathesis–stress model considered medication to be an important component and the families were sometimes seen as a core resource to help with maintaining adherence.[33 35 44 45] While the SYMPA model favoured a systemic understanding of the patient's situation, the importance of 'negotiation' with the patient regarding medication was emphasised.[30 31 37] Other systemic and postmodern-influenced models placed less emphasis on its use in the examined texts.[36 40] The greatest variation was in Open Dialogue, as patients were not given medication at the outset. If their condition did not improve, they were offered a low dose, with the aim to taper or discontinue its use over time.[32 60 61]

### System organisation

Finally, a major area of difference was the way the service was organised to deliver family involvement. Models based on the diathesis–stress theory usually required a small group of staff members to be trained as family involvement specialists who could manage the complexities of patient–family work.[33–35 43–45] Conversely, systemic models required whole teams to be trained in the principles of family involvement.[30–32 37–40 60 61] While specialist family training was still required, this was applied across the service and was not solely the responsibility of a smaller team.

**Table 3** Themes and subthemes relating to the role of patients and families in family involvement models

| Theme | Subthemes |
|---|---|
| 1. Families are a resource | |
| 2. Linear roles and relationships | 2.1. There is a 'patient' and a 'carer' |
| | 2.2. Families want to help |
| | 2.3. Family involvement is always beneficial |
| 3. Risk of identity loss | |
| 4. Implementation versus choice | |

## Role of patients and families

For the final research question, a thematic analysis[21] was conducted on the descriptions of the role of patients and families in the delivery of the models (summarised in table 3). The process was highly iterative and included multiple discussions with patients, family members and professionals. This analysis has also been supplemented by existing literature on this topic to reflect wider developments.

## Families are a resource

Families were conceptualised as a resource in a number of ways. They were often seen as 'potentially competent partners' (Gleeson *et al*[65] as cited in Seikkula *et al*[32]) in the stabilisation phase of the patient and in adherence to clinical procedures.[33 35 36 43 45] They were also perceived as a source of information about the patient's situation—whether directly or by observation.[30 31 33]

However, it was unclear whether family members were given the opportunity to refuse involvement while acknowledging the potential feelings of guilt that can emerge from this. Success of a model often seemed to depend on the willingness of the family to accept their 'helpful' or carer role and to engage with the techniques led by the professionals. Often, there were descriptions of how to engage unwilling family members.[32 33 45 60] This was addressed in some models[33 38 39] by emphasis on tools for family members to consider their own needs.

## Linear roles and relationships
### There is a 'patient' and a 'carer'

The relationship between the family and the individual accessing services was presented linearly, unless it was referring to codependency.[34] Even in systemic approaches that had circular causation as a theoretical reference, there was a clear, unidirectional carer and patient role.[30 31 36 37 40] The possibility of reciprocal support or a more egalitarian or independent relationship was not explicitly described in the examined models. This was also highlighted as an issue by individuals who hold mutual patient and carer roles during the analysis and consultation process. The role of reciprocity in caregiving in mental health is however explored in other research literature[66] and may hold a more prominent role in everyday clinical practice.

### Families want to help

Related to this, it was generally assumed that families either want involvement in the patient's treatment, or do not want it because they have been let down by professionals in the past.[33 38 39] This is in accordance with wider literature in the area, which highlights difficulties families experience when requesting involvement and information in clinical settings.[67 68] An alternative view that emerged from family members during the analysis process was that families might care for a relative but not want to feel responsible for their treatment. This was not explored in the examined texts but wider literature suggests a range of tools to support family members, which may help to overcome such difficulties in practice.[69]

### Family involvement is always beneficial

Moreover, while it was acknowledged that not everyone has supportive family members and that some relationships might be complicated, it was generally assumed that the involvement of family would be beneficial.[32 45 60] This idea is widely supported in wider literature.[1 2] However, as described next, the analysis process also indicated potential harms of involvement, which were often not explored in the examined texts. These ideas largely emerged from discussions of the personal lived experiences of patients and family members during the analysis process.

## Risk of identity loss

Considering the inherent vulnerability of being an individual in acute care, the positive and negative implications of involving others seemed greater. Clearly, family and friends could be a source of comfort and support in a difficult setting. However, sharing one's private information in a setting where they are the 'patient in need' could also risk altering their roles and relationships after they left acute treatment. In approaches with a wider social network approach, this potentially carried a higher risk. For example, colleagues may be invited to treatment meetings.[32 60 61] It seemed important to consider whether the patient in crisis could make a fully informed decision about the consequences of this and the possible impact this could have when they return to work.

Moreover, a general lack of accounts from patients themselves across the models meant that the patient could sometimes be described as a passive recipient of the interventions. Examples of individuals taking a more active role included psychoeducation that enlisted patients to share their own accounts[33] and patients being a core part of joint decision-making.[32 60 61]

## Implementation versus choice

Considering these points, we contemplated the role of patient choice in family involvement and how the structure of service organisation might affect this. If a whole service is set up to operate on the principles that family involvement is fundamental to psychiatric treatment, this makes it more likely to be implemented, as all staff are fully trained in facilitating it.[30–32 37–40 60 61] However,

depending on the delivery of the approach, this also has greater potential to weaken the patient's voice in the matter, making them feel pressured to involve others in a process they might have preferred to remain private. Conversely, if there are only specialist family involvement teams within a larger system, this can soon become an underused 'niche' service. In this case, the specialist team must rely on external factors such as managers and other colleagues seeing value in their approach, providing resources and collaborating to identify and refer 'suitable' families to the service, all of which can result in lower implementation.[13 33–35 43–45] Overall, while there may be no ideal approach to service organisation, the way 'choice' is presented appeared to be an important factor.

## DISCUSSION

This review broadly identified key family involvement models in acute mental health treatment and considered how their theoretical references are related to their delivery. From this, we considered how patients and families might be impacted. Despite major theoretical differences, we found many similarities in the components of the models, which raised the question: What is the intended aim when involving families and is it important to specify this? Namely, should all models be considered the same or is the theoretical basis an important aspect of delivery? Perhaps, as has been suggested for individual psychotherapy, non-specific factors (eg, the therapeutic relationship) determine the usefulness of the chosen model.[70] However, it may be important to place the model within its wider theoretical context to aid staff training and understanding, particularly in light of difficulties with implementation in this field.[3]

For example, there may be a conflict in how staff should conceptualise the patient and family roles and relationships. MacFarlane highlights how clinicians might struggle to be simultaneously family positive and inclusive despite being taught that they are at least partially responsible for the patient's problems.[71] Acknowledging these tensions and finding a way to integrate divergent world views may increase the likelihood of offering family involvement more consistently across services.

It is also important to consider that overall, a significant aspect of implementation is how well a model fits with a service's existing values.[72] Family psychoeducation is most strongly aligned with the existing biopsychosocial medical model, and this might be one reason why it fits more easily into existing services than postmodern or systemic models. The fundamentals of the latter approaches might be harder to train clinicians who have primarily been taught the biopsychosocial model, and to obtain resources through regular funding structures which prioritise medically focused clinical outcomes.

In considering implementation, the importance of patient and family experience should also be emphasised. For example, choice emerged as a significant consideration in our review. Too much focus on system-wide implementation could mean more likelihood of some patients feeling coerced into involving others during what is preferred to be a private process. This is of particular importance considering the potential harmful impacts of assuming individuals hold particular roles in their relationship. For example, a mother having a mental health crisis might not wish for her son to be present as a carer in her most acute phase, even if he is of adult age. This risks not only taking away her role as a person who holds authority and respect (a parent) but it might also impact on the dynamics of their relationship after the acute episode has subsided. This notion corresponds with Goffman's theories of the risk of identity loss in inpatient settings.[59]

Our review also indicated the potential for family members to feel disempowered by being viewed as a resource for services. This point was recently raised by Meijer and colleagues, who highlight the tension of being imposed the role of carer while having one's own goals and needs to attend to.[73] Rugkåsa's comprehensive review of elements of coercion in caring also makes the point that families may wish to be less involved in the care of their relative but fear the consequences of doing so.[74]

It is important to note here that a large body of research evidence supports the view that many families wish to have significantly more involvement, particularly in crisis contexts.[75 76] However, this review highlights the challenges of accommodating diverse needs in an already complex service setting. Overall, the offer of family involvement requires a delicate balance on the part of service leaders to support availability while maximising patients' and families' decision-making in the process.

### Strengths and limitations

This review has a number of strengths and limitations to consider. To our knowledge, this is the first conceptual review that has actively included people with lived experience of mental health services alongside academics and clinicians in the review team. This has allowed for in-depth integrations of personal and professional experiences of family involvement. This has contributed to a deeper understanding of the concepts present in each model and also what is *not* explained. From the latter, potential problems in the delivery of interventions were revealed. Research into patient safety emphasises the benefits of involving people with lived experience in identifying unknown latent harms.[77]

Although we conducted a systematic search, our findings are by no means an exhaustive list of all existing family involvement models. We avoided this for pragmatic reasons: in practice, the implementation of family involvement models can vary greatly, resulting in an infinite number of ways each component can be delivered. Investigating these infinite nuances will not necessarily lead to a better understanding of the field as a whole. Instead, a broad understanding of the diversity of the models that exist, and their common concepts, signifies the basis of family involvement in most settings.

Finally, due to the emphasis on published articles and manuals, the majority of literature was based in Europe, the USA and Australasia, or what is often referred to as the 'Western' medical system. This common problem can impact how relevant the results are cross-culturally.[78] Moreover, there may be an influence of the local context on the development and delivery of the included models, which might not translate to other settings. For example, rural environments might have more traditional family support structures than urban settings, affecting the nature of the involvement that can take place. However, when screening articles, we struck by how globalised family involvement approaches had become, in particular the adaptation of family psychoeducation to a number of international contexts.[79–81] How well this approach integrates with existing belief systems is a matter for another enquiry.

## CONCLUSIONS

Although family involvement models have been developed in the context of diverse theoretical perspectives and sociopolitical events, there are many commonalities in their components. Despite these commonalities, it must be acknowledged that the models are different in nature and underlying purpose. To enhance staff training and support implementation, there may be value in discussing the fundamentals of why family involvement is conducted, how it might be experienced by patients and families and how this relates to staff members' own perspectives. We therefore encourage further discussion of the differences and similarities between the various models and theories, taking into consideration different ideas about the nature of mental health and the purpose of involving families in these contexts.

**Twitter** @Ayse_D

**Acknowledgements** Thank you to all who generously contributed to the overall review process, particularly Paul McLaughlin (East London NHS Foundation Trust), SURF (the North London Service UserResearch Forum) and the SUGAR (Service User and carer Group Advising on Research)team.

**Contributors** AD designed the study, conducted searches and data extraction (in consultation with SS and SP), led on the analysis and prepared the manuscript. All authors contributed to the analysis, critically reviewed the paper and approved the final manuscript. SP and SS also provided overall guidance and supervision for the study.

**Funding** This article presents independent research funded by the National Institute for Health Research (NIHR), the East London NHS Foundation Trust and the Centre for Public Engagement (CPE) at Queen Mary University of London (QMUL). AD is funded by the NIHR Doctoral Research Fellowship (DRF-2015-08-071). DG was supported by the NIHR Collaboration for Leadership in Applied Health Research and Care (CLAHRC) North Thames at Barts Health NHS Trust. KB, GB and SC were supported by the Centre for Public Engagement at QMUL.

**Disclaimer** The views expressed are those of the authors and not necessarily those of the CPE, NHS, the NIHR, or the Department of Health.

**Competing interests** None declared.

**Provenance and peer review** Not commissioned; externally peer reviewed.

**Data sharing statement** There are no additional unpublished data to share.

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
