## [Reviewer comments · BMJ Open]

ARTICLE DETAILS

TITLE (PROVISIONAL)	Why Involve Families in Acute Mental Health Care? A Collaborative Conceptual Review
AUTHORS	Dirik, Aysegul; Sandhu, Sima; Giacco, Domenico; Barrett, Katherine; Bennison, Gerry; Collinson, Sue; Priebe, Stefan

VERSION 1 – REVIEW

REVIEWER	Professor Douglas MacInnes Canterbury Christ Church University United Kingdom I have worked previously with one of the authors (Stefan Priebe) on a funded project and we are currently part of a project team developing a NIHR funding proposal. I can confirm I have not had any discussion with Prof Priebe about this submission.
REVIEW RETURNED	27-May-2017

GENERAL COMMENTS	The paper describes a conceptual review of family involvement models used in non-specialist adult acute mental health settings. The importance of the topic is clearly described in the background section and there are references to relevant literature in the field. The authors have experience in this area from a clinical academic and user perspective with a wider consultation also involved. Some of the co-authors have previously published conceptual reviews. The review is clearly structured with signposts to readers on the approach to be taken though I think some of the procedures and areas explored could be expanded upon. I have noted these in my comments below. Main points Models not included The paper states on pg3 L58 that the authors “do not aim to present an exhaustive list of every existing family involvement model and set out to look at the distinct approaches that represented the diversity of the models that are used today”. I think this is a reasonable pragmatic approach. However, some models are not included such as the emancipatory model or the peer to peer support approach. It would be helpful for the readers of the article to get some idea some idea of the models that were not included in the review as the reasons for not including these. Cultural and Service Context On Pg 11 L8 The authors note “a significant aspect of implementation is how well a model sits with a services existing values”. In terms of the models identified. The majority of the models detailed in the paper are centred in a particular region of a country
---

(Somerset, Western Lapland, Plasir etc.). It would be helpful if some additional comments could be added to discuss the influence of specific cultural or service contexts in the development and maintenance of the models.

Thematic Analysis of Family and Patient Roles

In the thematic analysis pg10 and first part of pg11, the identified themes seem to focus either on the role of the family or patient the discussion examining the theme from both a patient and family perspective only apparent in the 2.1. "There is a patient and a carer" sub-theme and in a more limited sense there is some representation of both perspectives in the implementation vs choice theme. If it is the case that there are different themes being inferred for the patient role as opposed to the family role, this should be noted and the implications of this discussed. If not, it would be useful to try and examine the identified themes from both perspectives.

Level of agreement between different groups involved in the consultation

I was drawn to the comment on pg 12 L14 "potential for family members to be disempowered by being viewed as a resource for families". This is a reasonable point to make but a lot of the carer literature notes frustrations with their lack of involvement in the care and treatment of the patient and the lack of information they receive. It would be helpful to give an overview of the level of agreement between the people and groups involved in the consultation and the whether the identified themes were consistently agreed by all or if there were differences between different groups.

There are also some comments made in the paper where it I think it may be helpful to for further clarification:

Pg 4 L14 the chosen key models were finalised during the first stage of analysis. It would be useful to state what criteria governed these decisions.

Pg 7 L21. "Almost all of the modules routinely recommended medication along with a social intervention". It would be better to specifically state which models didn't recommended medication use (ie. Open Dialogue explicitly encourages no medication use for as long as possible). I'm aware the authors report this in the medication use section on page 9 but it would be useful to be specific about it here.

There seems to be a discrepancy between the information recorded in Table 2 and in the text. On pg 9 L54 the authors note "systemic models required whole teams to be trained in the principles of family involvement. This seems to at odds with the information presented in Table 2 where the models are recorded as "- relatively less or no emphasis" in the specialist teams/staff component column. My understanding of these approaches is that specialist training is required for staff working with the Calgary Family, Open Dialogue, SYMPA, and Somerset models.

It is noted on Pg 11 L9 family involvement has the potential to "weaken to the patient's voice in the matter making them feel pressurised to involve others in a process they might have preferred to remain private" and pg12 L3 "more likelihood of a patient feeling coerced into involving others". It would be useful to give some supportive evidence for this as an alternative view would be that having as many of a person's social network included in discussions both reduces the likelihood of a person withdrawing from their social network and also reduces stigmatisation thus helping the recovery process.

Figure 2 is a difficult to interpret. It would be useful to redraw this figure to clearly show where the various models are placed in

	relation to the various theories. An example of the uncertainty can be shown in relation to how Open Dialogue is denoted. In the diagram, Open is placed in the overlap between system theories and postmodern theories, while the D (of Dialogue) is placed in the overlap between postmodern theories and diathesis theories with the rest of the word (ialogue) in the postmodern theories circle. It's a little difficult to interpret from this where Open Dialogue sits in relation to the three theoretical concepts noted. It may also be easier for the reader to understand the relationship if the strength of these relationships are specifically stated in the text.
--	---

REVIEWER	Dr Grainne Fadden Meriden Family Programme Birmingham and Solihull Mental Health Trust Tall Trees Uffculme Centre Queensbridge Road Moseley Birmingham B13 8QY. I deliver a psychoeducational model of family work. I do not think this represents a conflict of interest but rather places me in a position to be able to comment on the accuracy of how these models have evolved over time compared to when they were developed in the 1980s. However, I felt I should mention this.
REVIEW RETURNED	04-Jun-2017

GENERAL COMMENTS	This paper is potentially useful and of interest in scientific terms in that there is confusion around different models of family work and what this means in terms of what clinicians should offer in clinical practice. The method used by the investigators in selecting 6 key models was appropriate and adequate I felt to explore this issue. However, I have some core issues with how some of the material and models are described in terms of accuracy. All of the models have evolved over time to keep pace with developments in thinking. The authors mention that there was 'an expert in family intervention' as one of the researchers, but do not specify which modality of family work this person was from. My main recommendation in order to ensure that the paper is accurate and complete is that the authors consult with those involved with the 6 models listed to check the accuracy of statements they are making relating to these models and how they are currently employed. Quoting papers that were written 20 to 30 years ago will not substitute for updated information on how these models have evolved and continue to evolve. This would not be difficult to do, as it is easy to make contact with, for example, Jaacko Seikkula in Finland, Frank Burbach in Somerset, the Meriden Family Programme in Birmingham (Falloon model) etc. I will elaborate on why I feel this is necessary and good practice, but basically I feel the authors and journal have a responsibility to ensure that models are not being misrepresented. Some sections of the paper come across as being 'back in time'. Specific examples:
---

Pg.7 Diathesis-stress model section - Expressed Emotion' is described as being the most widely accepted measure today, quoting references from 1972 and 1976. I would expect an analysis of where this is at currently, mentioning for example how findings have changed from earlier studies in the 70s to what the concept means e.g. relating to early psychosis, and the fact that its usage has been challenged because of the unhelpful labelling it often implies in relation to families, for example, families being described as 'overinvolved' when they are just being helpful.

Further down that paragraph, studies 32-34 which are systems approaches are quoted, although the section is on diathesis models, stating that they recommend medication and giving the impression that this is what psychoeducational models do.

Pg.9 - under Medication use - studies are quoted which are over 30 years old taking about the family's role in maintaining adherence to medication. These approaches have evolved to emphasise personal choice, different routes to recovery, and not asking family members to become surrogate staff, taking on roles they are not comfortable with.

Pg 9 System organisation - there is a statement that models based on diathesis-stress theory require a small group of staff to be trained as specialists. Again this does not keep pace with current developments. In 2016, Health Education England commissioned training for hundreds of front line clinical staff to develop these skills. The Meriden Family Programme has been training staff for 20 years and now have trained thousands of staff.

Pg.10 'Families are a resource' paragraph - this talks about families taking on caring roles, but does not describe detail of psychoeducational approaches where all individual family members are encouraged to set individual goals and look after their own needs. Numerous resources that have developed from these approaches reinforce this idea e.g. the Rethink 'Caring for Yourself' Manual and the Meriden MyCare App.

Under Linear roles and relationships 2.1 - reciprocal support is a central feature of family approaches in practice
2.2. Families not wanting to feel responsible for treatment - once again if the authors consult with those who continue to develop these approaches, they will discover this is commonly discussed and in fact families supported in not being responsible for another's treatment.

Pg. 11 Implementation and Choice - all family approaches are respectful of the individual choices of all family members, including the service user. It is inaccurate to suggest they do otherwise. They are also concerned for the wellbeing of all family members and their recovery.

In the Conclusion - there is discussion of coercion although family approaches are collaborative in nature. In general, it does not reflect how these approaches are practiced, for example, there is great sensitivity to roles, they are about empowering people rather than them being disempowered, they acknowledge that family members have their own needs.

	In summary, in order to present a balanced and accurate account, the views of those who develop and deliver these approaches should be taken account of in order to correct inaccuracies. The overall findings could then be looked at again to incorporate this, and would I feel then result in a really valuable paper that will be helpful both clinically and in conceptual and scientific terms.
--	--

REVIEWER	Scott Weich SchHARR, University of Sheffield United Kingdom
REVIEW RETURNED	05-Jun-2017

GENERAL COMMENTS	Thank you for allowing me to read and comment on this thoughtful and scholarly paper. The authors address a very important topic, and one for which evidence to guide clinical practice is sadly lacking. They have approached this subject in a meticulous way, and there is a great deal of worthy content here. My main criticism is that this is arguably too academic and insufficiently clear in what these findings might mean for patients and families (to use the term preferred by the authors), and for services. One acid test of this work is that it ought (at least in summary) be accessible to families of patients with serious mental illness. I think this paper as it stands would fail that test. I would raise the following further concerns:  1. The abstract is too discursive when it comes to the Results and Conclusions sections. I think all readers (even academic and scientific ones) would like to see some tangible examples of the findings of this research, rather than a description of what the authors did. Ditto in the conclusions: we need to see something specific that this research has discovered, and preferably something of relevance to patients, families and professionals. 2. The introduction presents (in one sentence) the view that involving families is beneficial. As the Discussion indicates, this is not always so. The introduction should include a more critical appraisal of the evidence of harms and benefits of family involvement, and perhaps a reflection on the ways in which these vary with context. As the authors suggest (but dismiss as historical curiosity), families are often the source of pathology through abuse and neglect. In fact, the most important risk factor for adult mental illness is being raised by a parent with mental illness, and this risk is thought to be mediated by parenting style. This must have some bearing on the benefits of involving families in the care of people with mental illness.  3. I found research question (4) - labelled (3) in my version of the manuscript - a bit curious. I would far rather have read a review of the effectiveness of different models of family involvement, and perhaps an evidence-based consideration of the different effects of their respective components. 4. The authors slip between referring to mental illness (generally) and schizophrenia (specifically), without explaining why.
---

	5. There is no mention of arguably the most widely used intervention for supporting families: the carers assessment. This is undertaken, and the results used, differently in different services. But a lot of resource is invested in this, and we don't know whether or how it helps. Many families reject the offer of this type of assessment, whose content may also vary between places. 7. Likewise, the biggest complaint voiced by families is being told by professionals that they are unable to answer questions of provide information about a family member "because of patient confidentiality". This related to one form of involvement that isn't mentioned here: being told about how a family member is doing, for instance after admission to hospital - much as one would if a loved one were taken ill and admitted to an acute hospital. Addressing this would help top ground the paper and allow readers to see a connection with their own experiences. 8. I didn't understand the use of the term 'postmodern', why it was chosen or how it differed from theories included under 'systems theories'. I suggest that the authors either revisit the choice of terminology or explain what they mean. 9. I think any review of this nature needs to explicitly address issues of culture and ethnicity, in so far as these influence experiences and family composition and functioning, and result in differences in involvement in the care of people with mental illness. It is well known, for example that preferences for involving the police when a family member is unwell varies between ethnic groups. 10. Finally, I struggled to see how the different elements of the review were being integrated, and this may be why I found it hard to have a clear sense of the study results or conclusions. I think this paper has a great deal to say, but it needs to offer readers some practical insights into optimal ways of involving (or indeed not involving) families of those being treated for mental illnesses.
--	--

VERSION 1 – AUTHOR RESPONSE

Comment:

The paper describes a conceptual review of family involvement models used in non-specialist adult acute mental health settings. The importance of the topic is clearly described in the background section and there are references to relevant literature in the field. The authors have experience in this area from a clinical academic and user perspective with a wider consultation also involved. Some of the co-authors have previously published conceptual reviews. The review is clearly structured with signposts to readers on the approach to be taken though I think some of the procedures and areas explored could be expanded upon. I have noted these in my comments below.

Response: Thank you for your comments and suggestions.

Comment:

Main points
Models not included

Comment:

The paper states on pg3 L58 that the authors “do not aim to present an exhaustive list of every existing family involvement model and set out to look at the distinct approaches that represented the diversity of the models that are used today”. I think this is a reasonable pragmatic approach. However, some models are not included such as the emancipatory model or the peer to peer support approach. It would be helpful for the readers of the article to get some idea of the models that were not included in the review as the reasons for not including these.

Response: Within our search strategy, we included all models that clearly fit the inclusion criteria e.g. (3) there needed to be a clear description of how families are involved in the patient’s treatment and (2) the model had to have the specific aim of managing an acute mental health situation or treating severe mental illness during the acute phase. This meant that we could only include models that were primarily focused on the involvement of family members to support the patient’s treatment. We feel it would be difficult to comment on other, similar approaches that did not meet this criteria within the limited word count. However, to address this and other comments, we have expanded the text to include variations in how the models might be delivered today, e.g. peer support is now mentioned as an example of how family psychoeducation has been developed to include other modalities.

Comment:

Cultural and Service Context

On Pg 11 L8 The authors note “a significant aspect of implementation is how well a model sits with a services existing values”. In terms of the models identified. The majority of the models detailed in the paper are centred in a particular region of a country (Somerset, Western Lapland, Plasir etc.). It would be helpful if some additional comments could be added to discuss the influence of specific cultural or service contexts in the development and maintenance of the models.

Response: We mentioned that the relevance beyond western contexts was limited in the discussion. Whilst we would only be speculating as to the reasons why, we have included a comment to state: “there may be an influence of the local context on the development and delivery of the included models, which might not translate to other settings. For example, rural environments might have more traditional family support structures than urban settings, affecting the nature of the involvement that can take place.”

Comment:

Thematic Analysis of Family and Patient Roles

In the thematic analysis pg10 and first part of pg11, the identified themes seem to focus either on the role of the family or patient the discussion examining the theme from both a patient and family perspective only apparent in the 2.1. “There is a patient and a carer” sub-theme and in a more limited sense there is some representation of both perspectives in the implementation vs choice theme. If it is the case that there are different themes being inferred for the patient role as opposed to the family role, this should be noted and the implications of this discussed. If not, it would be useful to try and examine the identified themes from both perspectives.

Response:

The thematic analysis process was highly iterative and included discussions with patients, carers and staff members in multiple mixed groups. Due to the limited word count, we did not originally emphasise themes that were already highly prevalent in the existing research literature e.g. carers’ desire for more involvement and information, especially as they were not commonly mentioned by people involved in this particular analysis process. However, as has been highlighted by other reviews too, there would be utility in incorporating these broader points to provide balance. Therefore, we have included perspectives from broader literature on family involvement and have clearly highlighted when this has been the case.

Comment:

Level of agreement between different groups involved in the consultation

I was drawn to the comment on pg 12 L14 “potential for family members to be disempowered by being viewed as a resource for families”. This is a reasonable point to make but a lot of the carer literature notes frustrations with their lack of involvement in the care and treatment of the patient and the lack of information they receive. It would be helpful to give an overview of the level of agreement between the people and groups involved in the consultation and the whether the identified themes were consistently agreed by all or if there were differences between different groups.

Response:

(As above) Additionally, the groups were all mixed and would not be possible to accurately attribute themes to particular groups. However, all the themes were discussed and agreed using a highly iterative and reflective discussion process. We hope to publish a separate manuscript which provides a detailed description this process.

Comment:

There are also some comments made in the paper where it I think it may be helpful to for further clarification:

Pg 4 L14 the chosen key models were finalised during the first stage of analysis. It would be useful to state what criteria governed these decisions.

Response: This is described in part 1 of the narrative synthesis (L39-44 on page 4), we have now included a signpost to this on Pg 4 L14.

Comment:

Pg 7 L21. “Almost all of the models routinely recommended medication along with a social intervention”.

It would be better to specifically state which models didn't recommended medication use (ie. Open Dialogue explicitly encourages no medication use for as long as possible). I'm aware the authors report this in the medication use section on page 9 but it would be useful to be specific about it here.

Response: Here, we wanted to emphasise that the majority of models have some elements of the diathesis-stress model by nature of taking place in a medical setting. We used to example of providing medication to address physical symptoms as a simple illustration of this, although other “medicalised” elements exist in all the models. We did not single out Open Dialogue as we did not interpret their approach as wholly anti-medication, but leaning more towards minimal use and tapering wherever possible. We expanded the point about medication in a separate section due to the very limited word count. We have however expanded the sentence on page 7 to include other elements of the diathesis-stress approach: “almost all of the models routinely recommended medication along with psychological support and social interventions”.

Comment:

There seems to be a discrepancy between the information recorded in Table 2 and in the text. On pg 9 L54 the authors note “systemic models required whole teams to be trained in the principles of family involvement. This seems to at odds with the information presented in Table 2 where the models are recorded as “– relatively less or no emphasis” in the specialist teams/staff component column. My understanding of these approaches is that specialist training is required for staff working with the Calgary Family, Open Dialogue, SYMPA, and Somerset models.

Response:

We meant to refer to the service approach rather than the training aspect. We broadly split the service approach into two options: (1) there is a specialist “family team” within the overall (generic) service or (2) all staff in the service have some aspect of family training, making it a whole system approach. We have added a sentence to clarify this: “Conversely, systemic models required whole teams to be trained in the principles of family involvement.[24,32-39] Whilst specialist family training was still required, this was applied across the service and was not solely the responsibility of a smaller team.”

Comment:

It is noted on Pg 11 L9 family involvement has the potential to “weaken to the patient’s voice in the matter making them feel pressurised to involve others in a process they might have preferred to remain private” and pg12 L3 “more likelihood of a patient feeling coerced into involving others”. It would be useful to give some supportive evidence for this as an alternative view would be that having as many of a person’s social network included in discussions both reduces the likelihood of a person withdrawing from their social network and also reduces stigmatisation thus helping the recovery process.

Response:

This emerged as a concern by individuals with lived experience in the review team. We then took all the emergent themes to service user and carer groups to check the wider relevance. The theme related to feelings of coercion resonated strongly with several people so we felt the need to include it. However, as described earlier, in the interests of providing balance we have updated the thematic analysis to include other recent literature.

Comment:

Figure 2 is a difficult to interpret. It would be useful to redraw this figure to clearly show where the various models are placed in relation to the various theories. An example of the uncertainty can be shown in relation to how Open Dialogue is denoted. In the diagram, Open is placed in the overlap between system theories and postmodern theories, while the D (of Dialogue) is placed in the overlap between postmodern theories and diathesis theories with the rest of the word (Dialogue) in the postmodern theories circle. It’s a little difficult to interpret from this where Open Dialogue sits in relation to the three theoretical concepts noted. It may also be easier for the reader to understand the relationship if the strength of these relationships are specifically stated in the text.

Response:

The figure was intentionally drawn to show that there aren’t many clear demarcations in the placement of the models within the theories. The slight overlap between sections was intentional to show this, but we understand that this is not very clear. We have moved the Open Dialogue section to a location that is close to the diathesis section but not overlapping to ease understanding. Whilst within the limited word count, it is not possible to explain all the reasons for the choices, we have included a sentence to state: “There were some overlaps in the theoretical references the models drew upon, as illustrated in figure 2” and have mentioned overlaps in other parts of the text. We have

also emailed authors asking about the placement of their model within the figure, but the only response we received did not indicate a need for change.

Comment:

This paper is potentially useful and of interest in scientific terms in that there is confusion around different models of family work and what this means in terms of what clinicians should offer in clinical practice.

Response:

Thank you, we hope the updated manuscript will better address these points.

Comment:

The method used by the investigators in selecting 6 key models was appropriate and adequate I felt to explore this issue. However, I have some core issues with how some of the material and models are described in terms of accuracy. All of the models have evolved over time to keep pace with developments in thinking. The authors mention that there was 'an expert in family intervention' as one of the researchers, but do not specify which modality of family work this person was from.

Response:

The expert in family intervention was Prof. Elizabeth Kuipers. We did not include her name as she was asked very specifically about her own model early on in the analysis process to clarify theoretical concepts (such as Expressed Emotion research), how they related to the development of the model, which other models could be considered "key" within this approach and how they differed from other approaches. This helped to give confidence in the theoretical references section and in the selection of the key papers for the family psychoeducation models. As Prof Kuipers was not consulted again on the content of the wider manuscript which was later developed with multiple people, we felt it not appropriate to include her name here, but we are very grateful for her time and expertise.

Comment:

My main recommendation in order to ensure that the paper is accurate and complete is that the authors consult with those involved with the 6 models listed to check the accuracy of statements they are making relating to these models and how they are currently employed. Quoting papers that were written 20 to 30 years ago will not substitute for updated information on how these models have evolved and continue to evolve. This would not be difficult to do, as it is easy to make contact with, for example, Jaacko Seikkula in Finland, Frank Burbach in Somerset, the Meriden Family Programme in Birmingham (Falloon model) etc. I will elaborate on why I feel this is necessary and good practice, but basically I feel the authors and journal have a responsibility to ensure that models are not being misrepresented. Some sections of the paper come across as being 'back in time'. Specific examples:

Response:

We have since contacted the mentioned authors, including Dr Fadden who has provided helpful resources on how the model is currently operationalised. In addition to this, we received a response from Prof. Lorraine Wright regarding the Calgary model. She stated:

“The interpretation of models of course is dependent on each author, practitioner, researcher, or educator. To comment on our how the models are implemented in practice would be extensive and beyond the scope of your paper.” Prof Wright also suggested changes to the wording of the description of the Calgary model, which have now been incorporated.

We have decided that based on the available information and within the scope of our analysis, it is not possible to fully and accurately describe how all the models are currently practiced. However, we can better describe that this is a review of fundamental, formative concepts that underpin key family involvement models and may be practiced in various ways by different services. Although we mentioned this in the conclusion (“in practice, the implementation of family involvement models can vary greatly, resulting in an infinite number of ways each component can be delivered”), we have also included information on page 7-8 which explicitly explains that current practice might differ from the original descriptions. We have also taken this approach to clarify and update other sections, including the thematic analysis.

Comment:

Pg.7 Diathesis-stress model section - Expressed Emotion' is described as being the most widely accepted measure today, quoting references from 1972 and 1976. I would expect an analysis of where this is at currently, mentioning for example how findings have changed from earlier studies in the 70s to what the concept means e.g. relating to early psychosis, and the fact that its usage has been challenged because of the unhelpful labelling it often implies in relation to families, for example, families being described as 'overinvolved' when they are just being helpful.

Response: The references from 1972 and 1976 are original descriptions of the EE approach, which remains a major basis of some types of family psychoeducation. We have now explicitly separated references to EE research concepts from other communication concepts to make clear that not all psychoeducation focuses on EE reduction.

Comment:

Further down that paragraph, studies 32-34 which are systems approaches are quoted, although the section is on diathesis models, stating that they recommend medication and giving the impression that this is what psychoeducational models do.

Response: The comment was not intended to single out medication, but made to address the idea that taking a systemic approach (for example) does not necessarily mean that the influence of the diathesis-stress model (which in this context is relatively “medical” in nature) is not present in the model. We have updated the sentence to include other elements of the diathesis-stress approach: “for example, almost all of the models routinely recommend medication along with psychological support and social interventions”

Comment:

Pg.9 - under Medication use - studies are quoted which are over 30 years old taking about the family's role in maintaining adherence to medication. These approaches have evolved to emphasise personal choice, different routes to recovery, and not asking family members to become surrogate staff, taking on roles they are not comfortable with.

Response:

As mentioned above we have now included information on page 7 which explicitly explains that current practice might differ from the original descriptions.

Comment:

Pg 9 System organisation - there is a statement that models based on diathesis-stress theory require a small group of staff to be trained as specialists. Again this does not keep pace with current developments. In 2016, Health Education England commissioned training for hundreds of front line clinical staff to develop these skills. The Meriden Family Programme has been training staff for 20 years and now have trained thousands of staff.

Response: (As above)

Comment:

Pg.10 'Families are a resource' paragraph - this talks about families taking on caring roles, but does not describe detail of psychoeducational approaches where all individual family members are encouraged to set individual goals and look after their own needs. Numerous resources that have developed from these approaches reinforce this idea e.g. the Rethink 'Caring for Yourself' Manual and the Meriden MyCare App.

Response:

As mentioned earlier, the thematic analysis section has now been supplemented with information from the wider literature to provide balance. We have explicitly stated when this has been the case.

Comment:

Under Linear roles and relationships 2.1 - reciprocal support is a central feature of family approaches in practice

Response: (As above)

Comment:

Families not wanting to feel responsible for treatment - once again if the authors consult with those who continue to develop these approaches, they will discover this is commonly discussed and in fact families supported in not being responsible for another's treatment.

Response: (As above)

Comment:

Pg. 11 Implementation and Choice - all family approaches are respectful of the individual choices of all family members, including the service user. It is inaccurate to suggest they do otherwise. They are also concerned for the wellbeing of all family members and their recovery.

Response:

This emerged as an important theme for patients and family members in various groups during the analysis process. We have included "depending on the delivery of the approach" to make clear that this point is dependent on delivery style rather than there being an inherent problem with any

particular model. With such great variation in the delivery of family involvement, we feel this is an important point to address to guide future clinical practice. We hope the amendment to the text clarifies this whilst retaining the original meaning of the theme.

Comment:

In the Conclusion - there is discussion of coercion although family approaches are collaborative in nature. In general, it does not reflect how these approaches are practiced, for example, there is great sensitivity to roles, they are about empowering people rather than them being disempowered, they acknowledge that family members have their own needs.

Response: As mentioned earlier, we have now updated several sections to clarify the results and include other perspectives.

Comment:

In summary, in order to present a balanced and accurate account, the views of those who develop and deliver these approaches should be taken account of in order to correct inaccuracies. The overall findings could then be looked at again to incorporate this, and would I feel then result in a really valuable paper that will be helpful both clinically and in conceptual and scientific terms.

Response: As above, thank you for your comments. We hope the revised manuscript is now clearer in its aims and presents more perspectives.

Comment:

Thank you for allowing me to read and comment on this thoughtful and scholarly paper. The authors address a very important topic, and one for which evidence to guide clinical practice is sadly lacking. They have approached this subject in a meticulous way, and there is a great deal of worthy content here.

My main criticism is that this is arguably too academic and insufficiently clear in what these findings might mean for patients and families (to use the term preferred by the authors), and for services. One acid test of this work is that it ought (at least in summary) be accessible to families of patients with serious mental illness. I think this paper as it stands would fail that test.

Response: Thank you for your comments.

Whilst we have made every effort to keep the language of the review plain for non-academic audiences, it is by nature a review of (sometimes quite abstract) theoretical concepts. The emergent findings of the review have been shared with service user and carer groups. If published, this will continue to be done in various formats (including plain English summaries and oral presentations) to ensure wider dissemination amongst patients, families and clinicians. We have also updated the abstract to provide more information in summary.

Comment:

I would raise the following further concerns:

The abstract is too discursive when it comes to the Results and Conclusions sections. I think all readers (even academic and scientific ones) would like to see tangible examples of the findings of this research, rather than a description of what the authors did. Ditto in the conclusions: we need to see something specific that this research has discovered, and preferably something of relevance to patients, families and professionals.

Response: The abstract has now been updated to include more findings.

We believe that the main potential of the review is to stimulate further discussion in this area, particular with regards to why staff training does not always translate to delivery in everyday practice. In terms of practical advice, we have now included information to state that the findings may be of particular relevance for staff training. For example, it might be important to place the chosen family involvement model within the context of their own world views and ideas about mental health to help increase implementation.

Comment:

The introduction presents (in one sentence) the view that involving families is beneficial. As the Discussion indicates, this is not always so. The introduction should include a more critical appraisal of the evidence of harms and benefits of family involvement, and perhaps a reflection on the ways in which these vary with context.

As the authors suggest (but dismiss as historical curiosity), families are often the source of pathology through abuse and neglect. In fact, the most important risk factor for adult mental illness is being raised by a parent with mental illness, and this risk is thought to be mediated by parenting style. This must have some bearing on the benefits of involving families in the care of people with mental illness.

Response: The role of families in causing harm is contentious, and it would not be possible to discuss this with any nuance within the very limited word count. For example, whilst parenting style or abuse within the family might be a risk, this does not necessarily diminish the benefits of working with families in acute settings. However, it also does not mean that it is always helpful to include families. For the purposes of this review, we wanted to illustrate the possibilities of (and reasons for) involving families in different ways, so the reader can reach their own conclusions in the context of their own setting and world view.

Comment:

I found research question (4) - labelled (3) in my version of the manuscript - a bit curious.

Response: Thank you for pointing this out, it has now been corrected.

Comment:

I would far rather have read a review of the effectiveness of different models of family involvement, and perhaps an evidence-based consideration of the different effects of their respective components.

Response:

The methodology of the conceptual review does not allow for synthesis of effectiveness. Multiple evidence reviews of family involvement models exist, although these are mostly confined to family psychoeducation as it is the most widely researched approach. A Cochrane review of Open Dialogue is also in progress: Pavlovic, R. Y., Pavlovic, A., & Donaldson, S. (2016). Open Dialogue for psychosis or severe mental illness. The Cochrane Library.

A separate point we wanted to make is that despite lots of existing evidence for the effectiveness of family involvement, there still seems to be a problem with translating this to everyday practice. We conducted the review to look at this long-standing issue from another angle.

Comment:

The authors slip between referring to mental illness (generally) and schizophrenia (specifically), without explaining why.

Response: We have now included information on page 5 to explain this.

Comment:

There is no mention of arguably the most widely used intervention for supporting families: the carers assessment. This is undertaken, and the results used, differently in different services. But a lot of resource is invested in this, and we don't know whether or how it helps. Many families reject the offer of this type of assessment, whose content may also vary between places.

Response:

Whilst the content varies, the main purpose of the carers assessment is to provide support for the carer themselves. Our inclusion criteria focused on interventions that involve family for the management/improvement of the patient's situation in acute treatment, rather than interventions that are primarily for supporting family members themselves.

Comment:

Likewise, the biggest complaint voiced by families is being told by professionals that they are unable to answer questions of provide information about a family member "because of patient confidentiality". This related to one form of involvement that isn't mentioned here: being told about how a family member is doing, for instance after admission to hospital - much as one would if a loved one were taken ill and admitted to an acute hospital. Addressing this would help top ground the paper and allow readers to see a connection with their own experiences.

Response:

This was not included as a stand-alone section as in our interpretation, it is not a specific intervention –it is something that could be said to be inherent in all of the models we have mentioned, as it is not possible to work with families in the described ways without providing information. Related to this, if family involvement were implemented more widely, fewer families would feel excluded in clinical settings due to reasons of confidentiality. We have now included several more mentions of the fact that many families wish to have more involvement in acute settings, which will hopefully be relatable to readers and add support to this point.

Comment:

I didn't understand the use of the term 'postmodern', why it was chosen or how it differed from theories included under 'systems theories'. I suggest that the authors either revisit the choice of terminology or explain what they mean.

Response:

The main difference we came across were the way postmodern and systemic theories conceptualised the idea of mental health. Postmodern theories tended to be more critical of "intervening" approaches

and focused more on the importance of working with uncertainty within the patient's context. Systemic approaches tended to be more focused on the importance of viewing the patient as part of many overarching systems and how these systems might maintain a certain state. Whilst there are many overlaps and the differences seemed subtle to us, we considered it important to still make this distinction and did our best to describe the concepts as we interpreted them within the limited word count. To help clarify this we have included additional information in the postmodern section. We also describe later that in practice, whilst there are different theories as to where the "problem" lies, in practice the family involvement models might be delivered in similar ways.

Comment:

I think any review of this nature needs to explicitly address issues of culture and ethnicity, in so far as these influence experiences and family composition and functioning, and result in differences in involvement in the care of people with mental illness. It is well known, for example that preferences for involving the police when a family member is unwell varies between ethnic groups.

Response:

The issue of culture and ethnicity is highly complex and we would not feel comfortable making inferences within this review using the chosen methodology, especially as we did not consult any individuals on matters of culture and ethnicity.

On a similar theme, we mentioned that the relevance beyond western contexts was limited in the discussion. Whilst we would only be speculating as to the reasons why, we have included a comment to state: "there may be an influence of the local context on the development and delivery of the included models, which might not translate to other settings. For example, rural environments might have more traditional family support structures than urban settings, affecting the nature of the involvement that can take place."

Comment:

Finally, I struggled to see how the different elements of the review were being integrated, and this may be why I found it hard to have a clear sense of the study results or conclusions. I think this paper has a great deal to say, but it needs to offer readers some practical insights into optimal ways of involving (or indeed not involving) families of those being treated for mental illnesses.

Response: Thank you for making this point, considering practical applications we have updated the conclusion:

"Although family involvement models have been developed in the context of diverse theoretical perspectives and socio-political events, there are many commonalities in their components. Despite these commonalities, it must be acknowledged that the models are different in nature and purpose. To enhance staff training and support implementation, there may be value in discussing the fundamentals of why family involvement is conducted, how it might be experienced by patients and families and how this relates to staff members' own perspectives. We therefore encourage further discussion of the differences and similarities between the various models and theories, taking into consideration different ideas about the nature of mental health and the purpose of involving families in these contexts."

VERSION 2 – REVIEW

REVIEWER	Professor Douglas MacInnes Canterbury Christ Church University United Kingdom
REVIEW RETURNED	06-Aug-2017

GENERAL COMMENTS	The authors have given a thorough responses to the points I raised. I also think these points have been addressed by the authors in the paper. The only exception is in relation to my first point that “It would be helpful for the readers of the article to get some idea some idea of the models that were not included in the review”. The authors reply given in their response to reviewers was that the aims of the review “meant that we could only include models that were primarily focused on the involvement of family members to support the patient’s treatment. We feel it would be difficult to comment on other, similar approaches that did not meet this criteria within the limited word count”. I think this reasonable justification would be helpful to be included in the article as it will inform readers why the no other models were examined.
--

REVIEWER	Dr Grainne Fadden Meriden Family Programme Birmingham and Solihull Mental Health NHS Trust Tall Trees, Uffculme Centre Queensbridge Road Birmingham B13 8QY United Kingdom
REVIEW RETURNED	17-Aug-2017

GENERAL COMMENTS	I am pleased to see that the comments of the reviewers have been taken on board and that care has been taken in addressing them. I think that the changes that have been made result in a more balanced and nuanced paper. I am happy that the issues raised have been addressed in a satisfactory manner. Thank you for being open to revising the paper - it is not easy and I appreciate the time that all of this takes. I feel this paper will be a significant asset to this area of research.
--

VERSION 2 – AUTHOR RESPONSE

We thank Prof MacInnes and Dr Fadden for their additional comments and are pleased to hear they were satisfied with our response. We have added a small section as recommended by Prof MacInnes. We have also corrected some minor formatting errors. We hope the changes will be found satisfactory by yourself and the reviewers. Our responses are as follows:

Prof MacInnes: “The authors have given a thorough responses to the points I raised. I also think these points have been addressed by the authors in the paper. The only exception is in relation to my first point that “It would be helpful for the readers of the article to get some idea some idea of the models that were not included in the review”. The authors reply given in their response to reviewers was that the aims of the review “meant that we could only include models that were primarily focused on the involvement of family members to support the patient’s treatment. We feel it would be difficult to

comment on other, similar approaches that did not meet this criteria within the limited word count". I think this reasonable justification would be helpful to be included in the article as it will inform readers why the no other models were examined".

Our response: Thank you again for your comments on the previous manuscript, we are pleased you feel they have now been addressed. We agree that it would be helpful to give an explanation of the limits of the studies that could be included in the review. As suggested, have now added the following text to the inclusion criteria section:

"The criteria meant that we could only include models where the primary focus was to involve family members in order to support the patient's care in acute settings. Whilst other approaches to involving families exist, they were considered to be beyond the scope of the current review."

Dr Fadden: "I am pleased to see that the comments of the reviewers have been taken on board and that care has been taken in addressing them. I think that the changes that have been made result in a more balanced and nuanced paper. I am happy that the issues raised have been addressed in a satisfactory manner.

Thank you for being open to revising the paper - it is not easy and I appreciate the time that all of this takes. I feel this paper will be a significant asset to this area of research."

Our response: Thank you again for taking the time to comment on our previous manuscript. We feel it has improved as a result of your suggestions, and the suggestions of other reviewers. We are pleased that the changes have addressed your previous concerns and thank you for being supportive.